# Asthma and Hyperglycemia: Exploring the Interconnected Pathways

**DOI:** 10.3390/diagnostics14171869

**Published:** 2024-08-26

**Authors:** Dharani K. Narendra, Sandhya Khurana

**Affiliations:** 1Baylor College of Medicine, Houston, TX 77030, USA; 2University of Rochester Medical Center, Rochester, NY 14620, USA

**Keywords:** asthma, hyperglycemia, metabolic syndrome, diabetes mellitus, insulin resistance, diabetes medications on asthma

## Abstract

The interplay between asthma and glucose metabolism disorders, such as hyperglycemia, has gained increasing attention due to the potential exacerbation of asthma symptoms and severity. This review explores the complex relationship between hyperglycemia and asthma, emphasizing the pathophysiological links, the impact of glucose metabolism disorders on asthma, and the effects of asthma medications on glucose levels. Hyperglycemia, often induced by asthma treatments like corticosteroids, has been associated with an increased risk of asthma exacerbations. This review delves into the pathophysiology underlying this association, highlighting the role of insulin resistance, metabolic syndrome, and obesity in both the development and management of asthma. Metabolic syndrome, characterized by abdominal obesity and hyperglycemia, independently increases the risk of worsening respiratory symptoms and asthma. Furthermore, this review examines the influence of various antidiabetic medications on asthma outcomes. Biguanides, like metformin, have shown promise in improving asthma outcomes in patients with type 2 diabetes mellitus and asthma. However, other medications have mixed results regarding their impact on asthma control and lung function. Considering these findings, this review advocates for further research into the role of metabolic pathways in asthma management. It calls for comparative studies and the inclusion of asthma-related outcomes in clinical trials of antidiabetic drugs to better understand their potential benefits for individuals with obesity and concurrent asthma.

## 1. Introduction

Asthma, a common respiratory disorder worldwide, is characterized by chronic inflammation of the airways. The Global Initiative for Asthma (GINA) defined asthma as a heterogenous disease “defined by the history of respiratory symptoms, such as wheeze, shortness of breath, chest tightness and cough, that vary over time and intensity, together with variable expiratory airflow limitation” [1]. The World Health Organization (WHO) estimates that over 300 million people worldwide suffer from asthma, and an additional 100 million are projected to be at risk [2]. In 2019, it was estimated that 37 million new cases were diagnosed worldwide, costing 21.6 million disability-adjusted life years (DALYs) [3]. Poor asthma control can have a significant impact on quality of life.

The relationship between asthma and hyperglycemia appears to be bidirectional. Glucose metabolism disorders are increasingly recognized as possible factors for asthma exacerbations and asthma severity. Studies indicate that hyperglycemia during childhood can heighten the likelihood of asthma development [4]. Furthermore, asthma treatments involving inhaled or oral corticosteroids may induce hyperglycemia as an unintended consequence. In individuals with asthma, hyperglycemia is associated with a heightened risk of exacerbation and severe symptoms. Here, we meticulously examine pivotal studies to gain a deeper insight into the intricate connection between hyperglycemia and asthma, while also exploring the implications and challenges arising from this association.

## 2. Pathophysiological Link between Asthma and Hyperglycemia

Glucose disorders stem from various causes and include the spectrum of silent hyperinsulinemia due to insulin resistance, stress hyperglycemia, impaired fasting glucose, impaired glucose tolerance, and diabetes mellitus [5]. These conditions negatively impact asthma, influencing its development and severity, and risk of exacerbations. Stress hyperglycemia has been defined as fasting glucose >6.9 mmol/L or random glucose >11·1 mmol/L without evidence of previous diabetes [6].

**Metabolic syndrome (MetS)** consists of clinical features like abdominal obesity, hyperglycemia, elevated triglycerides, hypertension, and low levels of high-density lipoprotein cholesterol [7,8]. More than one-fifth of the United States population and 60% of people with obesity suffer from MetS [9]. MetS is linked to the development of cardiovascular disease, diabetes mellitus, hepatitis steatosis, and cancer [10]. MetS is an independent risk factor for exacerbating respiratory symptoms, reducing lung function, causing pulmonary hypertension, and contributing to asthma [11]. Among MetS components, abdominal obesity has the strongest association with lung function impairment. A large cross-sectional study in a French adult population of 121, 965 showed that MetS was associated with greater lung function impairment (decrease in forced expiratory volume in one second (FEV1), forced vital capacity (FVC), [FEV1; OR 1.28; FVC; OR 1.41]) after adjusting for age, sex, smoking status, alcohol consumption, education, body mass index (BMI), physical activity, and cardiovascular disease [12]. Among all components of MetS, abdominal obesity was more strongly related to the outcome [12]. Asthma and metabolic syndrome synergistically decrease lung function [13]. In a large meta-analysis of 23,460 adults, obesity was associated with reduction in FEV, FVC, residual volume (RV), and total lung capacity (TLC) [14].

**Obesity:** Asthma and obesity are epidemiologically connected [15], and similar associations are observed with other markers of metabolic syndromes, such as insulin resistance and dyslipidemia [16]. Obesity is a predisposing factor for asthma both in adults and children. The derangements start in utero: Children whose mothers were obese during pregnancy or pre-pregnancy, or whose mothers exhibited gestational weight gain, have an increased risk of childhood asthma and wheeze [17]. Obesity-associated asthma is a heterogeneous disease encompassed by complex interrelationships among several mechanisms, including inflammatory processes, changes in immune response, and lung mechanics. Several asthma obesity phenotypes have been characterized using underlying disease processes [18]. Obesity may increase susceptibility to elevated type 2 helper (Th2) inflammation or a propensity for atopy. Both classic Th2-high and Th2-low inflammatory patterns are seen in obesity-associated asthma. The Th2-low asthma phenotype is associated with prominent respiratory symptoms and little eosinophilic airway inflammation [1]. There is evidence to show that asthma is more common in obese than lean individuals [18]. Obesity is a risk factor for developing asthma in adults and is a significant component of metabolic syndrome. A 2013 prospective cohort study followed 23,191 participants who were asthma-free at baseline from 1995 to 2008, with self-reported new asthma diagnoses at an average follow-up of 11 years [19]. Metabolic syndrome was identified as a risk factor for developing asthma (adjusted OR 1.57). This association remained consistent in sensitivity analysis using a stricter definition of asthma (adjusted OR 1.42). Among the components of metabolic syndrome, high waist circumference and elevated glucose or diabetes were linked to a higher risk of developing asthma in adulthood. There are two possible sequences of events: in one, obesity develops first, leading to changes that drive the development of asthma and affect its severity; in the other, asthma occurs first, and obesity develops afterward, possibly due to the effects of corticosteroid treatment and changes in physical activity. In addition, obesity could make asthma more difficult to control and treat. Obese patients tend to use more asthma maintenance medications and oral corticosteroids (OCSs), yet still experience poorer asthma control compared to their non-obese counterparts. Obesity-associated asthma patients have a high burden of exacerbations leading to emergency room visits and hospitalizations, including intensive care unit admissions [15,20,21]. Obese individuals often have other comorbidities that can negatively impact asthma, and obesity may influence the response to medications, likely contributing to the severity of the disease.

Mechanisms of interaction: Multiple factors are at play that contribute to the worsening of asthma severity and control in obese patients with asthma: the mere mechanical or physical changes from obesity, heightened systemic inflammation, sex hormones, arginine metabolism, components of metabolic syndrome, changes in microbiome, increased susceptibility to air pollution, and concomitant comorbidities [22].

Obesity is associated with higher expression and levels of inflammatory markers and adipokines in visceral fat. Among patients with asthma, an elevated BMI is associated with increased oxidative stress and increased levels of interleukin-6, interleukin-5, interleukin-13, and C-C chemokine receptors. Observational studies have reported impaired activity of natural killer cells, steroid-induced CD4+ T lymphocytes, and lower levels of surfactant protein B in obese asthma patients. As noted, both innate and adaptive immunity are key players in obesity with asthma.

Leptin and adiponectin are produced by adipose tissue that exerts metabolic effects on the lungs. Leptin levels are increased with obesity and are secreted in direct proportion to the adipose tissue mass. Leptins and leptin receptors are present in the airways of bronchial epithelial cells, and elevated leptin levels may modulate the immune reaction in the airways by inciting a robust pro-inflammatory response or cellular response towards the Th1 phenotype [23]. Leptin may also increase bronchial hyperactivity through its airway epithelial cell receptors. Airway reactivity has been shown to be strongly correlated with visceral fat leptin expression. Adiponectin is also expressed in airway epithelial cells; unlike leptin, adiponectin levels are decreased in patients with obesity and insulin resistance [24]. Certain genetic polymorphisms with leptin and adiponectin secretion (LEP and ADIPOQ genes) have been linked to the obesity-associated asthma phenotype [25].

The role of sex hormones in obesity-associated asthma has been studied. Gender-related differences in asthma incidence and changes during puberty have implicated the role of sex hormones [26].

The common comorbidities that affect obesity with asthma patients are obstructive sleep apnea (OSA), gastroesophageal reflux disease (GERD), mood disorders, and metabolic diseases.

**Insulin resistance** (IR): The term insulin resistance originated to mean “maladaptive in the setting of chronic overnutrition” [27]. Decreased cellular sensitivity to insulin is caused by improper fat accumulation in the liver. Initially, pancreatic beta cells secrete more insulin to preserve glucose homeostasis; however, with time, insulin production decreases, leading to diabetes [28]. IR is characterized by a reduced responsiveness to insulin in the liver, muscles, and adipose tissue. IR is inversely related to FEV1 and FVC in adolescents both with and without asthma [13]. A large study by Cardet et al. showed that IR significantly modified the association between obesity and current asthma [16]. In that study of 1241 patients (ages 18 to 25 years), the strength of association between obesity and asthma increased with increasing IR tertiles (OR 2.05). A similar increase in association with other components of metabolic syndrome, such as hypertriglyceridemia, hypertension, hyperglycemia, and systemic inflammation, was also observed [16]. This correlation highlights that targeting insulin resistance may be a therapeutic strategy for obese patients with asthma.

Mechanisms of Interaction: It is crucial to understand the effect of high doses of insulin on airway responsiveness. Hyperinsulinemia induces hyper-responsiveness of the parasympathetic nerves that control airway bronchoconstriction. A high insulin concentration increases neuronal acetylcholine release by disrupting the presynaptic inhibitory M2 muscarinic receptors on parasympathetic nerves. There is enhanced airway remodeling, characterized by increased collagen deposition in the lungs and the transition of epithelium to mucus-secreting cells. IR increases fibrosis, airway smooth muscle mass, and elevated expression of transforming growth factor-β1 (TGF-β1) [28]. Additionally, there is an increase in airway smooth muscle (ASM) contractility by heightened airway hyper-responsiveness, loss of inhibitory M2 muscarinic receptors, and enhanced vagally mediated bronchoconstriction. IR causes impairment in lung function, evident through reduced FEV1, FVC, and reduced forced expiratory flow over the middle half of the FVC (FEF25–75%) [28]. Moreover, IR increases the risk of respiratory tract bacterial colonization and increases the production of pro-inflammatory mediators from adipose tissue, such as interleukin -6 (IL-6) and tumor necrosis factor-alpha (TNF-α), leading to increased Th2 inflammation.

**Type 2 Diabetes Mellitus (T2DM)** is a frequent comorbidity in older patients with asthma, along with hypertension, coronary artery disease, and dyslipidemia. The prevalence of diabetes in asthma patients has increased [29,30]. Individuals with diabetes have a 2.2-fold higher risk of developing asthma compared to non-diabetics [20,31]. Poorly controlled diabetes has been associated with a higher risk of asthma [32]. Multi-morbidities are associated with increased hospitalization and healthcare costs [30]. The presence of pre-diabetes and diabetes is associated with higher rates of asthma exacerbation among obese adults with asthma. Wu and colleagues examined the influence of insulin resistance and metabolic syndrome on asthma morbidity [33]. That retrospective US cohort study of 5722 individuals found that higher hemoglobin A1C (HbA1C) levels were associated with higher asthma exacerbation rates. In the fully adjusted model, when compared to individuals with normal hemoglobin A1C, asthma exacerbation rates were 27% higher for HbA1C in the pre-diabetic range and 33% higher for HbA1C in the diabetic range [33]. Diabetes is also associated with an increased risk of asthma-related death during hospitalization for an asthma exacerbation [32]. Glucose disorders and metabolic syndrome significantly worsen asthma outcomes by increasing its incidence, severity, and exacerbation risk, particularly through mechanisms involving insulin resistance and obesity (see Table 1). Addressing these metabolic conditions, especially in obese individuals, could improve asthma management and reduce associated healthcare burdens.

Mechanisms of interaction: Chronic inflammation and increased pro-inflammatory cytokines are cardinal features of both asthma and diabetes mellitus. The receptor for advanced glycation end products (RAGE) is a multiligand receptor that has been shown to contribute to the pathogenesis of diabetes and asthma [34]. RAGE signaling is highly expressed in the lungs and induces chronic airway and vascular inflammation [35]. RAGE also has a regulatory role in T-cell proliferation and differentiation of both Th1 and Th2 cells. Systemic interleukin 6, a biomarker for metabolic dysfunction, is increased in patients with severe asthma and diabetes mellitus [36]. Matrix metalloprotein 9 mediates sputum overproduction due to airway epithelial barrier dysfunction caused by hyperglycemia, especially during asthma exacerbation [37] (Figure 1).

## 3. Effect of Asthma Medications on Glucose

Inhaled corticosteroids (ICS) and intranasal corticosteroids (INS) are the mainstay treatments for asthma. Multiple large observational studies suggest that high-dose ICS use is associated with an increased incidence of diabetes mellitus and worsening glycemic control [38]. Intranasal corticosteroid use is also associated with hyperglycemia [38]. Although ICS is targeted to provide localized therapy in the lungs, some of the dose is swallowed and absorbed through the gastrointestinal tract (oral bioavailability) and some through the lungs (pulmonary bioavailability). The blood concentration of an ICS relies on both absorption routes. Lower oral bioavailability is preferred to minimize systemic side effects. The oral bioavailability of current ICS varies from less than 1% for ciclesonide, mometasone, and fluticasone propionate to 26% for beclomethasone [39]. The administration of oral corticosteroids (OCSs) carries a dose-dependent risk of diabetes, an association observed even with limited exposure, such as four to five courses over a patient’s lifetime [1]. Long-term OCS therapies, independent of the dose, have been reported to elevate the risk of comorbidity and complications. Even very low doses of OCSs (<5 mg per day) can lead to complications. A focused literature review that included nine studies highlighted the dose–response relationship between OCS use in asthma and risk of metabolic derangements such as type 2 diabetes, hyperglycemia, obesity, dyslipidemia, and metabolic syndrome, thereby highlighting the importance of balancing the benefits of asthma control with OCS against the risk of side effects [39]. This led to the GINA guidelines recommending steroid-sparing therapies such as biologics over OCSs for uncontrolled severe persistent asthma.

Mechanisms of interaction: The glucocorticoids (GCs) absorbed from the respiratory system and the gastrointestinal tract are activated by 11b hydroxysteroid dehydrogenase type 1. GC binds to receptors on the liver cells, skeletal muscles, and adipose tissues. It exacerbates insulin resistance by increasing hepatic gluconeogenesis and decreasing skeletal muscle insulin sensitivity and glucose uptake. GCs upregulate angiopoietin-like 4 in adipocytes, which in turn increases the influx of non-esterified fatty acids (NEFA) and glycerol into hepatocytes, thus providing substrates for gluconeogenesis. Accumulation of NEFAs also reduces insulin release from the pancreatic beta cells [38].

## 4. Treatment Implications

One of the key steps in effective treatment of asthma is to address and optimally manage associated comorbidities. Asthma comorbidities impact both current asthma symptoms and future risk. Screening for diabetes and hypertension in patients on long-term treatment with oral or high-dose inhaled corticosteroids is important. In patients at risk of glucose metabolism disorders, in addition to optimizing management of comorbid conditions, efforts should focus on using the lowest effective dose of corticosteroids, including an ICS strategy when appropriate. In patients with severe or uncontrolled asthma, early introduction of advanced steroid-sparing therapies, including biologics, should be considered. For patients on high-dose ICS or maintenance OCSs, strategic de-escalation is imperative to mitigate the risk of hyperglycemia-related complications [38].

Diet, exercise, and weight loss play a vital role in managing obesity-associated asthma. Increased fruit and vegetable intake may decrease asthma development and exacerbation, while saturated fats, dairy intake, and meat products are associated with increased risk and exacerbating asthma symptoms [40]. A high-fiber diet and omega-3 fatty acids appear to be beneficial in observational studies.

Exercise-based interventions benefit patients with obesity-associated asthma. In a large observation study of 1.6 million, physical activity and exercise led to improved lung function [22].

Weight loss in obese individuals with asthma is associated with significant improvement in asthma symptoms and decrease in use of asthma medication [41,42]. Weight loss using dietary methods results in improvement in asthma severity, dyspnea, exercise tolerance, and acute exacerbations, including hospitalizations, due to asthma [43,44]. Furthermore, weight loss in obese asthmatics is associated with improvements in level of lung function and airway responsiveness to inhaled methacholine, without significant improvement in exhaled nitric oxide or other markers of type-2 airway inflammation [44]. Even a minimal amount of weight loss of >5% was associated with significant increase in FEV1 and FVC [45], and this is possible through virtual programs, in which >5% weight loss has better asthma control and quality of life [46].

## 5. Effect of Antidiabetic Medications on Asthma

As we have discussed the intricate, pathophysiological, and complex molecular links between asthma and metabolic syndrome, including diabetes mellitus, we can assume that there is a shared mechanism and potential for the use of antidiabetic drugs to treat airway inflammation. Currently, there is no FDA-approved treatment for asthma using antidiabetic drugs. We review the pivotal studies for each class and discuss the potential role of these drugs in asthma treatment.

Biguanides: Metformin is a prototype drug from the biguanide class. It is one of the first-line therapies for type 2 diabetes mellitus. Metformin improves insulin sensitivity through AMP-activated protein kinase in the liver, promoting fatty acid oxidation and inhibiting gluconeogenesis. Metformin has been shown to decrease the expression of pro-inflammatory mediators, including IL-6 and TNF alpha, and to inhibit nuclear factor-kB [47]. In a retrospective observational study by Wu et al. of 11,960 patients with asthma and type 2 diabetes mellitus, metformin use was associated with a lower risk of asthma exacerbation, asthma-related emergency department visits, and asthma-related hospitalization, but no significant difference in corticosteroid use [48]. Although mouse asthma models treated with metformin showed a decrease in lung tissue eosinophilic infiltration and pro-inflammatory cytokines, this has not been confirmed in observational studies in humans [48,49]. There are no data regarding the beneficial role of metformin in prospective controlled studies in asthma patients with and without obesity [28].

Sulfonylureas are most commonly used antidiabetic drugs. They attach to the receptors of pancreatic beta cells and increase insulin secretion. One retrospective observational cohort study found sulfonylureas provided modest protection in adults with T2DM against incident asthma, statistically equivalent to metformin initiation [50]. However, cardiovascular and hypoglycemic side effects limit the use of sulfonylureas in type 2 diabetes irrespective of asthma comorbidity [51].

Thiazolidines (TZDs) became unpopular in the last two decades due to side effects such as weight gain, peripheral edema, and skeletal fractures. Three randomized control trials of TZDs in asthma have reported no improvement in FEV1, asthma quality of life, or methacholine provocation concentration after 12–16 weeks of treatment [52,53,54]. The secondary outcomes in these studies revealed no change in peak expiratory flow rate, fractional exhaled nitric oxide (FENO), or symptom count.

Sodium-glucose co-transporter 2 channel inhibitors (SGLT-2i) prevent renal glucose reabsorption, leading to glucosuria and thus reducing hyperglycemia. In vitro studies have shown that SGLT-2i reversed inflammation by decreasing TNF receptor-1, IL-6, matrix metalloproteinase 7, and fibronection-1 levels. In a meta-analysis, Wang et al. reviewed 19 clinical trials and found that SGLT-2i use was associated with a significantly lower risk of asthma compared to placebo and DPP-4i [55]. The low incidence of asthma outcomes in both groups limits the validity of that study.

Glucagon-like peptide–1 (GLP-1) is a gut-derived hormone that improves insulin sensitivity and increases insulin secretion in response to oral food intake. GLP-1RA (GLP-1 receptor agonist) improves pancreatic beta cell function by promoting cell proliferation, stimulating insulin secretion, and inhibiting glucagon release. GLP-1 receptors are present in lung and immune cells. In addition, GLP-1RA promotes weight loss by suppressing food intake through early satiety and delayed gastric emptying. In a study of mouse asthma models, GLP-1A liraglutide decreased airway inflammation and mucus secretion [56]. In multiple studies, GLP-1A use is associated with improvement in asthma outcomes in adults with T2DM and comorbid asthma [57]. The combination of GLP-1RA with metformin, in contrast to metformin alone or metformin plus insulin, improved lung function (FEV1, FVC) in a prospective cohort of 32 patients with diabetes who did not have lung disease [58]. More extensive trials are needed to assess the true efficacy of this drug.

Dipeptidyl peptidase-4 inhibitors (DPP-4i): While GLP-1 stimulates insulin secretion from pancreatic beta cells in hyperglycemic states, it is cleaved by DPP-4, an adipokine released in excess amounts by adipose tissue in obese patients. By preventing inactivation of GLP-1, DPP-4 inhibitors significantly improve beta cell function and glycemic control. A meta-analysis revealed that DPP-4i did not reduce the risk of incident asthma relative to the placebo [55].

Insulin is often considered a final option in managing type 2 diabetes mellitus (T2DM), where it enhances glucose regulation and diminishes the risks of diabetes-associated complications and death. Several retrospective observational cohorts have shown a link between the use of insulin in T2DM and the onset of asthma [59]. Furthermore, in a prospective cohort study involving patients with type 2 DM without existing asthma or COPD (chronic obstructive pulmonary disease), the commencement of insulin therapy increased airway reactivity within the initial 60 days [60]. Inhaled insulin therapy was associated with increased airway responsiveness, airway smooth muscle proliferation, collagen deposition and peri-bronchial thickening, as seen in asthma patients [61].

Table 2 summarizes the therapeutic implications of hyperglycemia and asthma.

Table 3 summarizes the effect of antidiabetic drugs on asthma and their mechanism of action. Definitive evidence on the impact of antidiabetic medications on asthma is lacking, as most studies are retrospective observational. There is a need for more research focusing on metabolic pathways that could lead to potential improvements in asthma outcomes. Comparative studies and the integration of asthma-related outcomes in clinical evaluations of therapies for glycemic control are essential to determine their effectiveness in adults with glucose metabolism disorders and concurrent asthma.

## 6. Future Directions and Research Needs

The intricate interplay between asthma and hyperglycemia underscores a critical area of clinical and research interest, revealing the profound impact of metabolic disorders on respiratory health. In this review, we have attempted to highlight the significant associations between glucose metabolism disorders, such as hyperglycemia and insulin resistance, and asthma (see Figure 2). It has become evident that metabolic syndrome, characterized by obesity, hyperglycemia, and dyslipidemia, not only predisposes individuals to cardiovascular and metabolic diseases but also significantly influences asthma outcomes. The evidence presented underscores the need for a comprehensive approach to managing asthma, which includes careful consideration of metabolic health.

The effects of asthma medications on blood glucose levels further complicate the management of asthma in patients with concurrent metabolic disorders. While some antidiabetic medications, notably metformin, have shown potential to mitigate asthma exacerbations, the overall evidence highlights the complexity of treating asthma in the context of diabetes and obesity. The variability in responses to different classes of antidiabetic drugs calls for personalized treatment plans and underscores the necessity for ongoing research.

Future directions should focus on comprehensive studies that explore the mechanistic links between asthma and metabolic disorders, with an emphasis on developing therapeutic strategies that consider both comorbidities. There is a critical need for prospective, randomized controlled trials that investigate the effects of antidiabetic medications on asthma outcomes, aiming to integrate asthma endpoints in clinical trials of hypoglycemic therapies. Such endeavors will not only enhance our understanding of the bidirectional relationship between these conditions but will also pave the way for innovative, multidisciplinary approaches to treatment that optimize outcomes for patients with asthma and comorbid metabolic disorders.

## 7. Conclusions

The convergence of asthma and metabolic health challenges us to broaden our therapeutic lens, advocating for an integrative model of care that holistically addresses the needs of patients. By embracing this complexity, we can move towards more effective, personalized treatments that ensure better health outcomes for individuals afflicted by both asthma and glucose metabolism disorders.

## Figures and Tables

**Figure 1 diagnostics-14-01869-f001:**
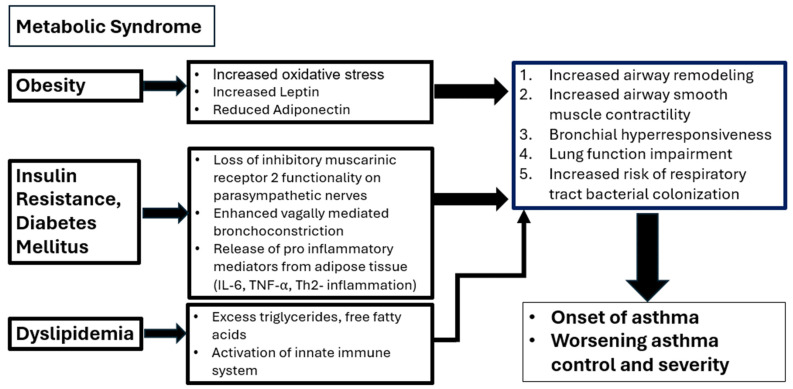
The relationship between the metabolic syndrome components, mechanism and their effect on asthma. Abbreviations: IL-6: Interleukin -6; TNF-⍺: Tumor Necrosis factor alpha; Th2: T helper cells type 2.

**Figure 2 diagnostics-14-01869-f002:**
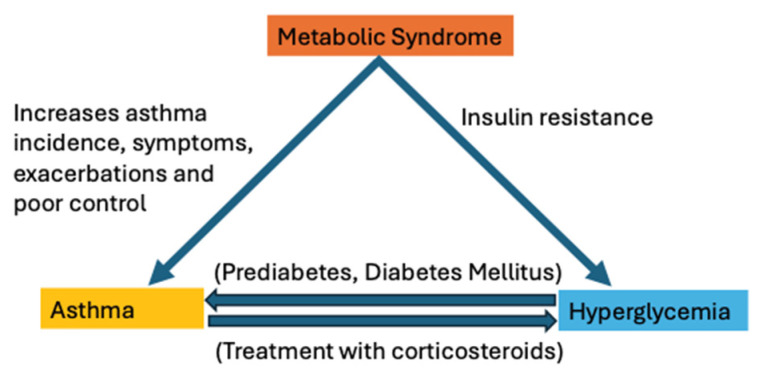
The interconnections between asthma, metabolic syndrome, and hyperglycemia.

**Table 1 diagnostics-14-01869-t001:** Potential links between Hyperglycemia and Asthma.

Metabolic syndrome, diabetes are risk factors for incident asthma
Asthma and metabolic syndrome synergistically decrease lung function
Pre-diabetes and diabetes were associated with high rates of asthma exacerbation among obese adults with asthma
RAGE * is a multiligand receptor that has been shown to contribute to the pathogenesis of diabetes and asthma
Increased systemic Interleukin 6, as a biomarker for metabolic dysfunction and inflammatory marker is seen in severe asthma patients with diabetes mellitus.
High glucose concentration increases the responsiveness of airway smooth muscles to contractile agents

* RAGE—receptor for advanced glycation end products.

**Table 2 diagnostics-14-01869-t002:** Therapeutic implications of hyperglycemia and asthma.

Recognize comorbidities in asthma (obesity, diabetes mellitus, gastroesophageal reflux disease, obstructive sleep apnea) and treat them
Diet, exercise, and weight loss are effective treatments in obesity associated asthma
Weight loss has shown to improve asthma symptoms and use of asthma medications
Recommending the use of non-corticosteroid inhalers and deescalating high dose inhaled corticosteroids when appropriate may mitigate risk of hyperglycemia
The variability in responses to different classes of antidiabetic drugs calls for personalized treatment plans and underscores the need for further research.

**Table 3 diagnostics-14-01869-t003:** The effect of antidiabetic drugs on asthma.

Antidiabetic Drugs	Mechanism of Action	Effects on Asthma
Metformin	Increases insulin sensitivity Decreases gluconeogenesis	Decreases the expression of proinflammatory markers (TNF-a, ROS, nitric oxide species) Decreases lung tissue eosinophilic infiltration Decreases asthma exacerbation
Sulfonylureas	Decreases insulin secretion	May lower risk of incident asthma
Thiazolidines (TZD)	Increases insulin sensitivity Decreases gluconeogenesis	No benefits in asthma
Sodium-glucose cotransporter-2 inhibitors (SGLT-2)	Decreases glucose reabsorption in kidneys	Decreases inflammatory markers Decrease risk of asthma
Glucagon like peptide 1 agonists (GLP-1A)	Increases insulin secretion Improves insulin sensitivity Inhibit glucagon release	Decrease airway inflammation, mucus secretion Improve asthma outcomes
Dipeptidyl peptodase-4 inhibitors (DPP4is)	Increases insulin secretion	No benefit
Insulin	Enhances glucose regulation	Increased bronchial smooth muscle proliferation and airway hyperresponsiveness Increase incidence of asthma

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
