# Peer review of "Asthma and Hyperglycemia: Exploring the Interconnected Pathways"

_diagnostics, 2024, doi:10.3390/diagnostics14171869_

Round 1

Reviewer 1 Report

Comments and Suggestions for Authors

It is a nice review about the interplay between asthma and hyperglycemia or better, the hyperglycemia-related diseases. It covers the most important diseases connected to increased blood glucose levels. However, the article needs some small corrections, as follows:

Check the entire paper for grammar and typing errors including the presentation of numbers and units. Add abbreviations when the term is used for the first time and then, use the abbreviation (like IL for interleukin)

Line 53 – check units

Both tables are not tables. Consider to change them to figures or as highlights at the end of the article or in conclusions.

Fig 1: Th2 cells – change to the proper name of “T helper cells type 2”.

Line 208: clarify the unit –“mg per day” would be enough.

Line 223; check to which degree ICS are absorbed from the respiratory system and correct where needed.

Line 304: add ‘agonist” after “receptor”.

Line 327: “asthma” and “hyperglycemia” should not be written in capital letters.

Otherwise, it is a nice article which can gain a lot of attention from the readers.

Comments on the Quality of English Language

There are only small typing errors, otherwise, it is easy to read and understand.

Author Response

We sincerely appreciate your thoughtful review and the time you took to provide valuable feedback on our manuscript. We have taken your suggestions seriously and have made the necessary corrections to address each of your comments. Below are the specific changes we have implemented in response to your feedback:  

  1. Grammar and Typographical Errors: We have thoroughly reviewed the entire manuscript for grammar and typographical errors, including the presentation of numbers and units. Additionally, we have ensured that abbreviations are introduced at their first mention and consistently used thereafter (e.g., "IL" for interleukin).  
  2.  Line 53 - Units: The units in line 53 have been double-checked and corrected as mmol/L
  3.   Both tables are not tables. Consider to change them to figures or as highlights at the end of the article or in conclusions: We have reviewed prior submissions to diagnostics and have noted similar context as tables instead of figures, hence have left them as tables.  
  4. Fig 1 - Th2 Cells: The term "Th2 cells" has been updated to "T helper cells type 2" in Figure 1 to ensure accuracy.  
  5.  Line 208 clarify the unit: The unit in line 208 has been clarified as “mg per day.”  
  6. Line 223 - check to which degree ICS are absorbed from the respiratory system and correct where needed: We have reviewed the degree of inhaled corticosteroids (ICS) absorption from the respiratory system and made the necessary corrections.
  7.   Line 304 - add ‘agonist” after “receptor”: The word “agonist” has been added after “receptor” in line 304.  
  8.  Line 327 - Capitalization: The terms “asthma” and “hyperglycemia” have been corrected to lowercase where appropriate.   We are confident that these revisions have improved the clarity and accuracy of the manuscript, and we are grateful for your insightful suggestions.

Reviewer 2 Report

Comments and Suggestions for Authors

The authors review asthma and hyperglycemia. My comments are as follows:

- The abstract appears to have more letters than allowed. It is repetitive.

- The text has different lettering styles.

- Item 2 would need to be rewritten. The MEtS section needs to be expanded. It repeats information in the asthma and obesity section. 

- Line 137 is missing reference.

- Images that better explain the different pathways involved.

- In Figure 1 it looks like everything is due to metabolic syndrome, the title could be asthma and its relationship to metabolic syndrome rather than hyperglycemia.

- In line 219 it says that there are 9 studies, put a table with them.

- In point 5 put the clinical trials in a table, summarize in a table the antidiabetic drugs with the targets and the effect on asthma.

- Figure 2 Improve quality

Author Response

Thank you for your detailed and constructive feedback on our manuscript. We appreciate your comments and have addressed each of your concerns with the following revisions:  

1. The abstract appears to have more letters than allowed. It is repetitive: We have revised the abstract to reduce the number of letters and eliminate any repetitive content, ensuring it is concise and to the point.  

2. The text has different lettering styles: The text has been standardized to a consistent lettering style throughout the manuscript.  

3. Item 2 would need to be rewritten. The MEtS section needs to be expanded. It repeats information in the asthma and obesity section: We have modified the section on Metabolic Syndrome (MEtS). Additionally, we have removed any repetitive information in the asthma and obesity sections to avoid redundancy.  

4. Line 137 - Missing Reference: A reference has been added to line 137  

5. Images that better explain the different pathways involved: this is has been explained in Figure 1

 6. In Figure 1 it looks like everything is due to metabolic syndrome, the title could be asthma and its relationship to metabolic syndrome rather than hyperglycemia: The title of Figure 1 has been revised to "Asthma and its Relationship to Metabolic Syndrome" to better reflect the content, as suggested.  

7. In line 219 it says that there are 9 studies, put a table with them: The table is beyond the scope of the article, however we highlighted the conclusion of the article.  

8. In point 5 put the clinical trials in a table, summarize in a table the antidiabetic drugs with the targets and the effect on asthma: We have added a table summarizing clinical trials, along with another table that lists antidiabetic drugs, their targets, and their effects on asthma.  

9. Figure 2  improve quality: The quality of Figure 2 has been improved to ensure better visual clarity.